# Transient Worsening of Dysphagia and Dysarthria after Treatment with Botulinum Toxin in Patients with Acquired Brain Injury

**DOI:** 10.3390/healthcare11243117

**Published:** 2023-12-07

**Authors:** Lucia Francesca Lucca, Luisa Spezzano, Francesco Bono, Maria Ursino, Antonio Cerasa, Francesco Piccione

**Affiliations:** 1S. Anna Institute, 88900 Crotone, Italy; l.lucca@isakr.it (L.F.L.); l.spezzano@isakr.it (L.S.); m.ursino@isakr.it (M.U.); 2Center for Botulinum Toxin Therapy, Neurology Unit, A.O.U. Mater Domini, 88100 Catanzaro, Italy; f.bono@unicz.it; 3Institute for Biomedical Research and Innovation (IRIB), National Research Council of Italy (CNR), 98164 Messina, Italy; 4Pharmacotechnology Documentation and Transfer Unit, Preclinical and Translational Pharmacology, Department of Pharmacy, Health and Nutritional Sciences, University of Calabria, 87036 Rende, Italy; 5Unit of Neurorehabilitation, Padua Hospital, University of Padua, 35122 Padova, Italy; fpiccione61@gmail.com

**Keywords:** botulinum toxin, spasticity, dysarthria, dysphagia, acquired brain injury

## Abstract

Although botulinum toxin is widely considered an effective and safe treatment for a variety of neurological conditions (such as disabling spasticity), local or systemic adverse effects have often been reported. This study describes three cases of patients with severe acquired brain injury who were receiving speech therapy for recovering dysphagia and dysarthria but showed worsening of these symptoms after receiving BoNT treatment for motor spasticity. To increase clinicians’ knowledge of these adverse effects, we present our cases and explore their significance to avoid major complications such as aspiration pneumonia.

## 1. Introduction

Spasticity is increased muscle tone caused by a neurological insult, resulting in a lesion of the upper motor neurons, and occurs in conditions such as stroke, multiple sclerosis, cerebral palsy, and traumatic brain and spinal cord injury [1]. Pathological muscle hyperactivity may be focal, multifocal, segmental, multisegmental, or generalized and, if untreated, may develop into disabling spasticity [2], which is perceived by the affected person or caregivers as an impediment to body function, activities, and/or participation [3]. Spasticity is typically treated with rehabilitation procedures [4,5,6] in conjunction with pharmacological therapy [7,8,9]. However, these techniques frequently do not ensure adequate outcomes.

Therefore, other treatments, such as infiltration with botulinum toxin and the implantation of a baclofen pump, are proposed. The European Advisory Board has recently developed an algorithm that can assist clinicians in choosing a treatment for disabling spasticity with a baclofen pump and/or botulin toxin [2]. Several systematic reviews supported the conclusion that botulinum toxin A (BoNT) is a safe and effective treatment for a variety of neurological conditions, such as disabling spasticity [10,11]. Generally, BoNT is injected directly into muscles and causes inhibition of the release of acetylcholine at the neuromuscular junction. Intramuscular injections for the treatment of spasticity are administered into skeletal muscles, which can vary in size from small foot and hand muscles to large muscles such as the quadriceps and gastro–soleus complex, which can require injections at multiple sites in the same muscle. Current therapeutic guidelines recommend treatment with BoNT for focal upper and lower extremity spasticity [11,12], whereas the treatment of multifocal or multisegmental upper or lower extremity spasticity [13] requires higher total doses per session and may increase the risk of systemic diffusion with the development of clinically visible adverse effects.

The administration of BoNT therapy for spasticity is not without potential issues. In the last ten years, many adverse effects have been linked to BoNT treatment in neurology patients, which has prompted a rigorous analysis of their nature, occurrence, and clinical significance [14]. Early detection of side effects is mandatory in order to guarantee the security and effectiveness of BoNT therapy as well as to maximize patient treatment. Adverse effects associated with BoNT include symptoms in different physiological systems such as respiratory, neurological, musculoskeletal, cardiovascular, visual, oropharyngeal, bowel, bladder, and immune symptoms [15,16,17,18,19]. Among these, dysarthria and dysphagia have been reported as possible side effects [19]. Dysarthria is reported especially when treating spasticity in or around the orofacial muscles, such as the muscles controlling facial expression or mastication [14,15,16]. The incidence of dysarthria varies depending on several factors, including the dosage of BoNT administered, the specific muscles targeted, and individual patient characteristics. Another known side effect of BoNT therapy for spasticity is dysphagia, which can also arise depending on the underlying neurological condition being treated as well as the dosage, place, and kind of BoNT administered [18,19].

In this study, we describe three cases of patients with vascular and traumatic severe acquired brain injuries who were receiving speech therapy for recovering dysphagia and dysarthria but showed worsening of these symptoms after receiving BoNT treatment for spasticity.

## 2. Materials and Methods

### Case Series

Three patients with acquired brain injury, including one woman (case 1) and two men (cases 2 and 3), were clinically monitored at the Intensive Rehabilitation Unit (IRU) in Crotone, Italy. At admission, all patients underwent clinical (Coma Recovery Scale-Revised, CRS-r;) and logopedic assessments, which were repeated at 1 and 4 weeks after BoNT treatment. The examination was performed by two clinicians with experience in disorders of consciousness who were blind to any other result. Clinical information and therapy schedule are shown in Table 1. During logopedic assessment, all patients underwent clinical evaluation for dysphagia and dysarthria using the Gugging Swallowing Screen (GUSS) scale [20] and the Dysarthria Assessment Robertson Profile (DP) and Self-Assessment Questionnaire [21]. GUSS consists of two parts: the indirect swallowing test and the direct swallowing test. The higher final the score, the better the performance. The total score ranges from 0 to 20 and is subdivided as follows: 0 to 9 means that the pretest or ingestion of semisolid food failed and the dysphagia is classified as severe with a high risk of aspiration; 10 to 14 means that the patient swallows semisolid food without difficulty but has difficulty with liquids and the dysphagia is classified as moderate with a low risk of aspiration; 15 to 19 means there was no difficulty in swallowing semisolid or liquid foods, while there was difficulty with solid foods, so dysphagia is rated as mild with minimal risk of aspiration; and score 20 corresponds to the absence of difficulty, so mild/no dysphagia with minimal risk of aspiration was noted. The revised DP assesses the quality of respiration, phonation, facial muscles, diadochokinesis, reflexes, articulation, intelligibility, and prosody for a total 71 assessment items. A score from 0 to 4 is assigned as follows: 0 absent, 1 poor, 2 moderate, 3 good, 4 excellent.

BoNT injection dose was applied in the three patients according to international guidelines and systematic review [22,23] by one of the authors (F.B.). All medical and demographic information of the three patients is reported below.

## 3. Results

### 3.1. Case N° 1

The first case is a 33-year-old female patient, hospitalized in the emergency for coma post hemorrhage (GCS = 3) due to the rupture of an arteriovenous malformation (AVM) in the left cerebellar hemisphere. She was immediately intubated, mechanically ventilated, and treated with an external ventricular shunt for hydrocephalus. She was subsequently tracheostomized and treated with a ventriculoperitoneal shunt (GCS score = 3).

Approximately four months after the brain injury, the patient was transferred from the Intensive Care (ICU) to the Intensive Rehabilitation Unit (IRU) at S. Anna Institute in Crotone, Italy. She was admitted in a minimally conscious state (CRS-r score = 10), was autonomously breathing via a tracheostomy tube, was fed by PEG, and was totally dependent in all activities of daily living. During the neurorehabilitation stay, she had ventriculoperitoneal shunt malfunction, then was transferred to neurosurgery for shunt replacement. At the second admission to the IRU, she presented a new cerebral hemorrhage for which, nine months after the first event, she underwent cerebellar AVM embolization. After the resolution of the various complications with transfers to the neurosurgery department, about ten months after the first bleeding event, the preliminary speech therapy assessment revealed a GUSS score of 0/20 and a Dysarthria Profile-revised score of 25/284 (Table 2 and Table 3).

Almost a year after the event, the patient underwent treatment with BoNT for perioral spasms and sialorrhea. She was treated with injections of onabotulinumtoxin A (total dose 40 U) in the parotid gland, submandibular gland, masseter, mental, orbicularis, and elevator of the upper lip muscles on both sides. During the week following treatment, worsening of both dysphagia and dysarthria was observed. The post-inoculation speech therapy evaluation revealed a worsening of the dysarthria characterized by forced phonation due to hyperadduction of the vocal cords, slow and pulling articulation with poor intelligibility, increased drooling, strangled voice with reduced speed and low pitch, and hypernasalization: pre-inoculation GUSS 5/20, Profile Dysarthria Score 121/284, post-inoculation GUSS 2/20, Profile Dysarthria Score 58/284 (Table 2 and Table 3). The patient underwent sessions of intensive speech therapy, including the administration of homogenous boluses, exercises in front of the mirror, dry swallowing and saliva control, stimulation of the veil and thermal stimulation, respiratory exercises for coordination pneumophonics, vocalizations, and facial massage. The recovery of the pre-inoculation performance level was noticed after 3 weeks of intensive logopedic treatment (Table 2 and Table 3).

### 3.2. Case N° 2

The second case is a 13-year-old boy, a victim of a traffic accident as a pedestrian hit by a car (GCS = 4), with a diffuse axonal injury of the third degree. After forty days in the ICU, he was transferred to the IRU in a vegetative state (CRS-r = 4), with autonomous respiration via a tracheostomy tube and was fed via a nasogastric tube. During the stay in neurorehabilitation facility, frequent attacks of Paroxysmal Sympathetic Hyperactivity (PSH) with tachycardia, tachypnea, hypertension, hyperthermia, profuse sweating, and a pathological pattern in decerebration occurred. The successive neuropsychological evaluation detected the emergence from the vegetative state (presence of functional communication) (Table 1) and the first logopedic evaluation registered scores of GUSS 15/20 and Dysarthria Profile-revised 58/284 (Table 2 and Table 3). Approximately four months after the trauma, the patient showed a significant reduction in PSH attacks but multisegmental spasticity.

Abobotulinumtoxin A (total dose 1000 U) was administered in the left sternocleidomastoid muscle and in the semispinalis, splenius capitis, pectoralis major, biceps brachii, brachioradialis, flexor digitorum superficialis and profundus, adductor hamstring, gastrocnemius, soleus, tibialis posterior, and extensor hallucis longus of the right side. During the week following the inoculation, speech therapy evaluation showed lingual hypotonia, slowing down of labio-bucco-facial movements, hypernasalization with inadequate veil control, and involuntary movements of the chin and lips.

Clinical scales recorded pre and post inoculation were as follows: GUSS: 17/20, Dysarthria Profile-revised score: 204/284 and GUSS·13/20, Dysarthria Profile-revised score: 131/284, respectively (Table 2 and Table 3). No further adverse related events—muscle weakness, oculomotor deficits, respiratory failure, bowel or bladder or infections—were detected. After the toxin treatment, the patient underwent several daily sessions of speech therapy, together with exercises of visual feedback, dry swallowing, stimulation of the palatal veil and thermal stimulation, pneumophonic coordination, vocalizations, and relaxation and lengthening of the buccinator and orbicularis muscles of the mouth. The recovery of the pre-inoculation performance level was noticed after three weeks of intensive logopedic treatment (Table 2 and Table 3).

### 3.3. Case N° 3

The third case is a 29-year-old male patient with ischemia ponto-mesencephalic (GCS = 3) secondary to the dissection of the right vertebral artery until the basilar artery. Cerebral panangiography was performed and a stent was placed. He was admitted to the ICU, mechanically ventilated, tracheostomized, and underwent gastrostomy tube placement. Approximately one month later, he was transferred to the IRU after he emerged from a coma with severe neuromotor, phono-articulatory, and swallowing deficits. The first logopedic assessment showed the following scores: GUSS 0/20, Dysarthria Profile-revised score 23/284 (Table 2 and Table 3). For widespread spasticity, the patient underwent onabotulinumtoxin A (total dose 1200 U) three months after the event. The pectoralis, biceps brachii, pronator teres and quadratus, superficial and deep flexor digitorum, quadriceps femoris, medial and lateral gastrocnemius, and soleus were treated on the right side, whereas on the left side the pectoralis, biceps brachii, pronator teres and quadratus, superficial and deep flexor digital rum, biceps femoris, adductor hamstring, medial and lateral gastrocnemius, soleus, and extensor pedis were treated.

The speech evaluation one week after the injection of the toxin highlighted the loss of the phonemes recovered during the previous treatment, slowing of labio-bucco facial movements, an increase in drooling, characteristic voice (“potato in the mouth”), slowing of chewing phases, and worsening of dysphagia for liquids. The scores at the assessment before and after the toxin treatment were GUSS 10/20, Dysarthria Profile-revised 171/284 and GUSS 7/20, Dysarthria Profile-revised 101/284, respectively (Table 2 and Table 3). The patient reported no other adverse events, such as muscle weakness, oculomotor deficits, respiratory problems, bowel or bladder problems, or infections. The recovery of the pre-inoculation performance level was noticed after 20 days of intensive logopedic treatment (Table 2 and Table 3).

## 4. Discussion

In this study, we present three clinical cases of patients with severe acquired brain injuries (two with vascular origin and one with traumatic origin). All patients were admitted to the ICU in a coma with a GCS score lower than eight, requiring a tracheal cannula and a PEG implant for feeding. All patients during the neurorehabilitation stay emerged from the altered consciousness state. Treatment with BoNT was indicated in every patient for treating spasticity. At the time of BoNT treatment, all patients underwent neuromotor treatment and speech therapy for recovering dysphagia and dysarthria. During the week after BoNT treatment, patients experienced worsening dysphagia and dysarthria, as shown by GUSS and DP scales. No patients experienced other adverse events such as muscle weakness, oculomotor deficits, respiratory failure, bladder-/bowel-related events, or infections. They resumed intensive speech therapy and recovered their pre-inoculation performance levels within twenty days. Despite the various injection sites and toxin doses, in all three cases, dysphagia and dysarthria worsening were temporary side effects. In the literature, toxin treatment has been shown to be effective in treating dysphagia by chemically denervating the cricopharyngeal muscle [24,25,26,27,28]. Furthermore, the injection of BoNT into the thyroarytenoid muscles improves symptoms in patients with spastic dysarthria [29]. However, dysphagia and dysarthria are also reported in the literature as possible adverse events of BoNT treatment for spasticity. In particular, after examining the clinical records of toxin exposure from September 1992 to March 2009 as reported by the U.S. Food and Drug Administration (FDA), it was found that neurological events occurred in 25.3% of treatments, and the most common side effects of BoNT were dry mouth, hoarseness, and dysphagia, which were largely linked to aspiration pneumonia [30].

The data reported in our case series study agree with this evidence. Indeed, in the first case, the patient was treated with onabotulinumtoxin A (total dose 40 U) at the level of the salivary glands and peri-buccal muscles. For this reason, a local spread of the toxin leading to a worsening of dysphagia and dysarthria could be hypothesized. Our findings agree with Van Hulst et al. [31], who reported 33% of the adverse effects on oral motor function after initial injections of BoNT-A into the submandibular glands to treat excessive drooling in children with central nervous system disorders. Similarly, Phadke et al. [13] reported 272 oropharyngeal adverse events following treatment with onabotulinutoxin A, including the following: dysarthria, dysphagia, pharyngitis, eating disorders, speech disorders, tongue paralysis/bleeding, chewing disorders, abnormal saliva production, dysphonia, laryngospasm, vocal cord disorders, lockjaw, drooling, ageusia, stomatitis, aphagia, dry mouth, oral swelling, oral pain, tongue pain, toothache, and slow speech. Moreover, in agreement with Punga et al. [32], serious side effects, such as anaphylaxis, dysphagia, respiratory insufficiency, and widespread muscle weakness, have been documented hours to weeks after starting BoNT therapy in facial muscles. These uncommon systemic effects, however, are only observed in people with underlying medical problems or at extremely high dosages. In a recent systematic review, Heikel et al. [33] evaluated the impact of BoNT treatment in the management of pediatric sialorrhea. Adverse occurrences included speech problems, dysphagia, and dysarthria ranging from mild to moderate. Finally, in two people with cerebral palsy who underwent BoNT treatment to the parotid and submandibular glands, adverse effects, such as increased salivary viscosity and dysphagia and dysarthria, were noted [34].

In the second case, the treatment of neck muscles could explain the side effects, which occurred 1 week after injection but disappeared after 3 weeks. In this case, the treatment was proposed to avoid the implantation of a baclofen pump in a young individual who showed a positive recovery. A case history of eighteen patients treated with a toxin for cervical dystonia revealed similar timings in symptoms [35]. In total, 11% of the patients had dysphagia prior to receiving botulinum toxin treatment; following treatment, these patients’ dysphagia symptoms persisted, but 33% of them had new dysphagic symptoms. The negative effects started to show up about five days after the injection and disappeared approximately 2 weeks later.

Finally, in the third case, although the baclofen pump system represented the best indication, following the family’s preference we adopted a less invasive BonT intervention [36]. In this case, we believe that the worsening of symptoms could be related to the dosage level. Indeed, the patient underwent onabotulinumtoxin A (total dose 1200 U) treatment for widespread multisegmental and multifocal spasticity. Our findings agree with previous findings [11] demonstrating that intervention for multifocal or multisegmental spasticity requiring a high dosage may increase the risk of adverse effects. As claimed by the Royal College of Physicians [37], when large doses are applied to the neck or proximal upper limb, dysphagia mostly develops. However, it is important to keep in mind that those who have had a stroke or brain damage may have compromised swallowing reflexes. Patients with a history of dysphagia should be carefully injected with higher doses of BoNT-A, particularly if they do not have percutaneous gastrostomy (PEG) feeding tubes.

Despite the fact that BoNT treatment for spasticity is generally recommended in multidisciplinary rehabilitation programs of brain injury patients [38], the reported adverse effects should be considered in order to adopt some useful strategies to avoid other major complications. For instance, according to patient and caregiver preferences, the first-line option [2] for patients with multifocal and multisegmental spasticity could be to replace the pump implant with a hypothetically less invasive and effective treatment that is still reversible (see patient N° 3). It is important to consider dysphagia as a potential side effect and to put protective measures in place during the first week following botulinum toxin therapy for patients with severe neurological compression, such as those with ABI. If the patient is fed only through a PEG tube (cases 1 and 3) or changes the consistency of their meal (case 2), they should be given intravenous or PEG hydration only. Healthcare personnel must use the utmost caution due to the possibility of ab ingestis pneumonia along with worsening dysphagia. Conversely, a patient with dysarthria may experience discomfort, but without potentially fatal consequences. The speech therapist overseeing the patient in this situation might let the patient and their family members know about this possibility and reassure them that it might only be a temporary deterioration. In this instance, it is advised to intensify speech therapy and, if at all feasible, involve family members [39].

## 5. Conclusions

The purpose of this study is to inform and alert healthcare professionals to the possibility of potential adverse effects in the clinical management of patients with severe acquired brain damage who receive BoNT therapy for spasticity. In fact, all the reported clinical cases showed a (transient) worsening of dysphagia and dysarthria after the treatment with botulinum. These unexpected negative symptoms might be caused by poorly understood mechanisms related to the toxin’s injection. However, the real causality is difficult to establish because there are no easily accessible tissue or blood indicators that can link the toxin directly to the adverse effects.

Generally speaking, in order to take precautions against potentially more dangerous or fatal events (i.e., aspiration pneumonia), understanding potential adverse consequences resulting from the administration of the toxin is recommended. For instance, one can follow the seven-point clinical protocol for safer neurotoxic usage of BoNT presented by Hristova et al. [30].

## Figures and Tables

**Table 1 healthcare-11-03117-t001:** Timing of treatments and clinical evaluations.

Case	Admission ICU	GCS Score ICU	Admission IRU	CRS-r at Admission IRU	Emersion from VS	First Logopedic Assessment	Logopedic Assessment before BoNT	BoNT Intervention	Logopedic Assessment 1 Week after BoNT	Logopedic Assessment 4 Weeks after BoNT
1	07/01/14	3	08/04/14	10	10/10/14	16/10/14	10/12/14	11/12/14	18/12/14	07/01/15
2	29/10/13	4	12/11/13	4	24/02/14	26/02/14	02/04/14	03/04/14	10/04/14	07/05/14
3	24/03/16	3	28/04/16	18	28/04/23	08/05/16	06/07/16	07/07/16	14/07/16	06/08/16

ICU: Intensive Care Unit; IRU: Intensive Rehabilitation Unit; GCS: Glasgow Coma Scale; CRS-r: Coma-Recovery Scale-revised; VS: vegetative state; BoNT: botulinum toxin.

**Table 2 healthcare-11-03117-t002:** GUSS scores.

Case 1				Case 2				Case 3			
Admission	Pre-BoNT Treatment	Post-BoNT Treatment	Discharge	Admission	Pre-BoNT Treatment	Post-BoNT Treatment	Discharge	Admission	Pre-BoNT Treatment	Post-BoNT Treatment	Discharge
0/20	5/20	2/20	7/20	15/20	17/20	13/20	20/20	0/20	10/20	7/20	18/20

BoNT: Botulinum toxin injection.

**Table 3 healthcare-11-03117-t003:** Dysarthria Profile (DP-Revised scale)—scoring form.

	Task (See Norms Sheet)		Case 1				Case 2				Case 3			
			I DP	Pre	Post	Dis	I DP	Pre	Post	Dis	I DP	Pre	Post	Dis
**I RESPIRATION**	**1 Ability to sustain/s/on exhalation also//(N)**		1	2	1	2	1	3	2	3	0	2	1	3
	2 Ability to “crescendo” on/s/or//		0	2	1	2	0	3	2	1	0	2	1	3
	3 Ability to “diminuendo” on/s/or//		0	2	1	2	0	3	2	1	0	2	1	3
	4 Ability to repeat series of /s/or//		0	2	1	2	0	3	2	1	0	2	1	3
	5 Ability to synchronise respiration with phonation		1	3	1	2	1	3	2	2	0	2	1	3
	Sub- total score		**2**	**11**	**5**	**10**	**2**	**15**	**10**	**8**	**0**	**10**	**5**	**15**
**II PHONATION**	1 Ability to initiate /a:/		1	3	2	2	2	3	2	4	0	3	3	3
	2 Ability to sustain /a:/ (N))		1	2	1	2	2	3	2	4	0	2	1	3
	3 Ability to say /a:/ very loudly		1	2	1	2	2	3	2	4	0	2	1	3
	4 Ability to “crescendo” on a/a:/ <		1	2	1	2	1	3	2	3	0	2	1	3
	5 Ability to “diminuendo” on a/a:/ >		1	2	1	2	1	3	2	3	0	2	1	3
	6 Ability to repeat series of /a:/		1	2	1	2	1	3	2	3	0	2	1	3
	7 Ability to raise pitch on /a:/ (3 pitches)		1	2	1	2	1	3	2	2	0	2	1	1
	8 Ability to lower pitch on /a:/ (3 piches)		0	0	0	1	1	3	2	1	0	2	1	1
	9 Ability to glide up scale on /a:/ (octave)		0	0	0	1	1	3	2	1	0	2	1	1
	10 Ability to glide down scale on /a:/ (octave		0	0	0	1	1	3	2	1	0	2	1	1
	11 Ability to maintain adequate volume in speech		1	2	1	2	1	3	2	2	0	2	1	2
	12 Quality of voice		1	2	1	2	1	3	2	1	0	2	1	2
	Sub total score		**9**	**19**	**10**	**21**	**15**	**36**	**24**	**29**	**0**	**25**	**14**	**26**
**III FACIAL MUSCULATURE**	1 Change expression to a smile	Face	0	2	1	2	1	3	2	3	1	3	2	4
	2 Change expression to a frown		0	2	1	2	1	3	2	3	1	3	2	3
	3 Open and close mouth	Lips	1	2	1	3	1	3	2	3	1	3	2	3
	4 Purse lips “oo” /u:/		1	2	1	3	1	3	2	3	1	3	2	3
	5 Stretch lips “ee” /i:/		1	2	1	3	1	3	2	3	1	3	2	3
	6 Maintain lip closure (hold a spatula against resistance)		0	1	0	2	1	3	2	3	1	2	1	3
	7 Protrude tongue	Jaw	0	2	1	2	1	3	2	4	1	2	1	4
	8 Retract tongue		0	2	1	2	1	3	2	2	1	2	1	3
	9 Lateral tongue movement (against spatula resistance)		0	2	1	2	1	3	2	2	1	2	1	3
	10 Pass tongue over teeth	Tongue	1	2	1	2	1	3	2	3	1	2	1	3
	11 Tongue tip into Right cheek (against resistance)		0	0	0	2	1	3	2	3	1	2	1	3
	12 Tongue tip Left cheek (against spatula resistance)		1	2	1	2	1	3	2	3	1	2	1	3
	13 Raise tongue tip to the alveolar ridge		1	2	1	2	1	3	2	3	1	2	1	3
	14 Elevate soft palate /a:/		1	2	1	2	1	3	2	3	1	2	1	3
	15 Elevate soft palate on a series of /a:/		1	2	1	1	1	3	2	3	1	2	1	3
	16 Cough as strongly as possible		1	2	1	1	1	3	2	3	1	2	1	3
	17 Raise tip tongue in mouth		0	0	1	1	1	2	1	3	1	2	1	3
	18 Raise tongue tip out of mouth		0	0	1	1	1	2	1	3	1	2	1	3
	19 Elevation veil of palate on /a/	Veil	1	2	1	2	0	2	1	2	1	2	1	3
	20 Elevation veil of palate on a series of /a/		0	1	1	2	0	2	1	2	1	2	1	3
	Sub total score		**10**	**32**	**18**	**39**	**18**	**56**	**36**	**57**	**20**	**45**	**25**	**62**
**IV DIADOCHOKINESIS**	1 Open and close the mouth rapidly (N)		1	2	1	2	1	3	2	3	1	3	2	3
	2 Protrude and retract the lips quickly (N)		0	2	1	2	1	3	2	3	1	3	2	3
	3 Protrude and retract the mouth rapidly (N)		0	2	1	2	1	3	2	3	1	3	2	3
	4 Raise and lower the tip of the tongue quickly (N)		0	2	1	2	1	3	2	3	0	3	2	3
	5 Move the tongue from side to side quickly		1	2	1	2	1	3	2	3	0	3	2	3
	6 Ability to repeat “u-i” rapidly (N)		0	1	0	2	1	3	2	3	0	3	2	4
	7 Ability to repeat “pa-pa” rapidly (N)		0	1	0	2	1	3	2	3	0	2	1	4
	8 Ability to repeat “ta-ta” rapidly (N)		0	1	0	2	1	3	2	3	0	2	1	4
	9 Ability to repeat “ka-ka” rapidly (N)		0	1	0	2	1	3	2	3	0	2	1	4
	10 Ability to repeat “ka-la” rapidly (N)		0	1	0	2	0	3	2	2	0	2	1	4
	11 Ability to repeat “p-t-k” rapidly (N)		0	1	0	2	0	3	2	2	0	2	1	4
	Sub total score		**2**	**16**	**5**	**22**	**9**	**33**	**22**	**31**	**3**	**28**	**17**	**39**
**V REFLES**	1 Chewing		0	2	1	1	0	3	2	3	0	2	1	3
	2 Swallowing solid foods		0	2	1	1	0	3	2	3	0	2	1	3
	3 Swallowing liquids		0	2	1	1	0	3	2	3	0	2	1	3
	4 Inhibit sialorrhea at rest		0	2	1	1	0	3	2	3	0	3	2	3
	5 Inhibit sialorrhea during feeding		0	2	1	1	0	3	2	3	0	3	2	3
	6 Inhibit sialorrhea in conversation		0	1	0	1	0	3	2	3	0	3	2	3
	7 Coughing and scraping		0	2	1	1	0	3	2	3	0	3	2	3
	Sub total score		**0**	**13**	**6**	**7**	**0**	**21**	**14**	**21**	**0**	**18**	**11**	**21**
**VI ARTICULATION**	1 Ability to repeat initial consonants		1	2	1	2	1	4	2	3	0	3	2	4
	2 Accuracy of vowel sounds		1	2	1	2	1	4	2	3	0	3	2	4
	3 Ability to repeat consonants clusters		0	1	0	1	1	2	1	3	0	3	2	4
	4 Ability to repeat polysyllabic words		0	1	0	1	1	2	1	3	0	3	2	4
	5 Ability to repeat sentences		0	2	1	1	1	2	1	4	0	3	2	4
	Sub total score		**2**	**8**	**3**	**7**	**5**	**14**	**7**	**16**	**0**	**15**	**10**	**20**
**VII INTELLIBILITY**	1 Reading intelligibility /therapist)		0	2	1	2	0	3	2	2	0	3	2	3
	2 Reading intelligibility (family members)		0	3	2	2	1	2	1	3	0	3	2	4
	3 Reading intelligibility (strangers)		0	2	1	2	1	2	1	2	0	3	2	2
	4 Speech intelligibility (therapist)		0	2	1	2	0	3	2	2	0	2	1	3
	5 Speech intelligibility (family members)		0	3	2	2	1	2	1	3	0	2	1	4
	6 Speech intelligibility (strangers)		0	2	1	2	1	2	1	2	0	2	1	2
	Sub totascorel		**0**	**14**	**8**	**12**	**4**	**14**	**8**	**14**	**0**	**15**	**9**	**18**
**VIII PROSODIY**	1 Maintaining adequate speed of speech		0	2	1	2	1	3	2	3	0	3	2	3
	2 Increased speed of speech		0	2	1	2	1	3	2	3	0	3	2	3
	3 Maintaining an adequate pace		0	2	1	2	1	3	2	3	0	3	2	3
	4 Use of proper intonation		0	1	0	2	1	3	2	2	0	3	2	3
	5 Imitation of different accent modes		0	1	0	2	1	3	2	2	0	3	2	3
	Sub total score		**0**	**8**	**3**	**10**	**5**	**15**	**10**	**13**	**0**	**15**	**10**	**15**
	**TOTAL SCORE**		**25**	**121**	**58**	**128**	**58**	**204**	**131**	**189**	**23**	**171**	**101**	**216**

**DP:** First Dysarthria Profile score; **Pre:** Dysarthria Profile-revised score pre BoNT inoculation; **Post**: Dysarthria Profile-revised score post BoNT inoculation; Dis: discharge.

## Data Availability

Data are contained within the article.

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
