# Peer review of "Transient Worsening of Dysphagia and Dysarthria after Treatment with Botulinum Toxin in Patients with Acquired Brain Injury"

_healthcare, 2023, doi:10.3390/healthcare11243117_

Round 1

Reviewer 1 Report

Comments and Suggestions for Authors

Thank you very much for letting me review this very interesting article “Transient Worsening of Dysphagia and Dysarthria After Treatment with Botulinum Toxin in Patients with Acquired Brain Injury.

The article in general is well written. Some issues however need to be addressed by the authors.

The references added to this article should be updated, there are numerous recent studies available related with the use of botulinum toxin for treatment of Dysphagia and Dysarthria, which could be cited.

Also, there are several studies which shows the beneficial effect of use Botulinum for the treatment of Dysphagia and Dysarthria after neurological disease. So, it needs to be discussed why treatment with botulinum toxin leads to worsening situation in these 3 cases.

Also, three different doses of botulinum were given in 3 cases, it needs to be discussed in detail what was the basis of selecting these 3 different doses.

Author Response

Thank you very much for letting me review this very interesting article “Transient Worsening of Dysphagia and Dysarthria After Treatment with Botulinum Toxin in Patients with Acquired Brain Injury. The article in general is well written. Some issues however need to be addressed by the authors.

  1. The references added to this article should be updated, there are numerous recent studies available related with the use of botulinum toxin for treatment of Dysphagia and Dysarthria, which could be cited.

REPLY: Following the reviewer’s suggestion, we included a lot of papers about the use of botulinum toxin for the treatment of Dysphagia and Dysarthria

  1. Also, there are several studies which shows the beneficial effect of use Botulinum for the treatment of Dysphagia and Dysarthria after neurological disease. So, it needs to be discussed why treatment with botulinum toxin leads to worsening situation in these 3 cases.

REPLY: Following all the reviewers’ suggestions, the discussion and conclusion have been widely re-formulated to better explain our data.

  1. Also, three different doses of botulinum were given in 3 cases, it needs to be discussed in detail what was the basis of selecting these 3 different doses.

REPLY: Following the reviewer’s suggestion, we included several references about the use of botulinum toxin for the treatment of Dysphagia and Dysarthria

Reviewer 2 Report

Comments and Suggestions for Authors

Dear authors, despite the worthy effort of evaluation, the manuscript presents critical issues of both structure and content which I report here:

1) the introduction is poorly represented, with only 9 notes (it deserves to be expanded);

2) in the introduction, after note 4, the comment is not supported by any note (it must be added);

3) from the introduction you must separate the last sentence and create the section dedicated to the aims and objectives of the research (separate and expand the sections);

4) the materials and methods section does not present a clear distinction between the materials and methods used in detail and the scales used must be specified in detail and also indicated in the tables reported in the manuscript, otherwise the scores cannot be understood (separated, expanded and corrected the text);

5) in point 2.1 after note 11 there are no more references (they must be added);

6) point 3 should be the results section and not "case reports" rather than the type of study (correct);

7) in the discussions the possible contributing causes that could exclude the negative effect of the therapeutic use of BoNT-A are omitted and there are no statistical analyses to support the conclusions (review the discussions, add the supporting statistical analysis, and evaluate the co-clinical factors that play in favor and against);

8) the conclusions present reference no. 14 which should be inserted either in the introduction or in the discussions and explained (make changes);

9) the conclusions are not supported by statistical analyses, e.g. multiple regression (insert statistical analysis to support the conclusions);

10) the total number of notes should be, at least, at least 30-40, of which 50% at least from the last three years, and written according to the journal's editorial rules (they must be expanded and revised).

For these reasons I will propose more changes without rejection, to respect your work done so far. Good work!

Author Response

Dear authors, despite the worthy effort of evaluation, the manuscript presents critical issues of both structure and content which I report here:

  • the introduction is poorly represented, with only 9 notes (it deserves to be expanded);

REPLY: Following the reviewer’s suggestion, references and evidence supporting our claims have been included

2) in the introduction, after note 4, the comment is not supported by any note (it must be added);

REPLY: Following the reviewer’s suggestion, references and evidence supporting our claims have been included

3) from the introduction you must separate the last sentence and create the section dedicated to the aims and objectives of the research (separate and expand the sections);ù

REPLY: Done

4) the materials and methods section does not present a clear distinction between the materials and methods used in detail and the scales used must be specified in detail and also indicated in the tables reported in the manuscript, otherwise the scores cannot be understood (separated, expanded and corrected the text);

REPLY: This section has been reformulated following the reviewer’s suggestion

5) in point 2.1 after note 11 there are no more references (they must be added);

REPLY: Please consider that this section has been reformulated. However, in the previous description of the clinical scale used for assessing dysphagia and dysarthria, we briefly described the GUSS and Dysarthria Assessment Robertson Profile. Thus, additional references are not required.

6) point 3 should be the results section and not "case reports" rather than the type of study (correct);

REPLY: Done.

7) in the discussions the possible contributing causes that could exclude the negative effect of the therapeutic use of BoNT-A are omitted and there are no statistical analyses to support the conclusions (review the discussions, add the supporting statistical analysis, and evaluate the co-clinical factors that play in favor and against);

REPLY: Discussion has been widely re-formulated following the comments of reviewer n°3.

8) the conclusions present reference no. 14 which should be inserted either in the introduction or in the discussions and explained (make changes);

REPLY: Done

9) the conclusions are not supported by statistical analyses, e.g. multiple regression (insert statistical analysis to support the conclusions);

REPLY: We agree with this reviewer, but clinical assessment recorded in only three patients (as usual in a case series study) prevents us from the opportunity to perform parametric or not parametrical statistical evaluation of the clinical improvement/worsening at follow-up.

10) the total number of notes should be, at least, at least 30-40, of which 50% at least from the last three years, and written according to the journal's editorial rules (they must be expanded and revised).

REPLY: Since the paper has widely been re-formulated a large number of new references have been added.

Reviewer 3 Report

Comments and Suggestions for Authors

The case reports presented in this study discuss patients with severe acquired brain injuries who underwent Botulinum toxin (BoNT) therapy for spasticity, which led to temporary exacerbation of dysphagia and dysarthria. While the authors have provided insightful clinical observations, there are certain areas that warrant further attention and elucidation. To improve the quality and significance of the case reports, it is recommended that the authors consider addressing the suggestions outlined below. 

·     The Introduction could benefit from improved flow and organization. It might be helpful to separate the general information about spasticity and BoNT treatment from the introduction of the cases to ensure a smooth transition between these elements.

·       In the Introduction paragraph mentions that patients experienced "worsening of symptoms" after BoNT treatment, but it does not specify what these symptoms were. Providing a brief overview of the specific symptoms that worsened would make the context clearer.

·       While the case reports note potential reasons for the observed adverse effects (e.g., local toxin spread, dosage levels), the discussion lacks an in-depth exploration of the mechanisms underlying these adverse effects. Providing more insights into the pathophysiological basis of the observed symptoms would strengthen the report.

·       The case reports focus on the immediate post-treatment period. Including information about the patients' long-term outcomes and whether any additional treatments or interventions were required for symptom resolution would provide a more comprehensive picture.

·       In the discussion, mention potential alternative treatments or strategies to mitigate the adverse effects of BoNT treatment in patients with severe acquired brain injuries. Are there strategies to avoid or manage these complications?

·       Discuss the clinical implications of these findings. What can clinicians learn from these cases to improve patient care or make more informed treatment decisions?

Author Response

The case reports presented in this study discuss patients with severe acquired brain injuries who underwent Botulinum toxin (BoNT) therapy for spasticity, which led to temporary exacerbation of dysphagia and dysarthria. While the authors have provided insightful clinical observations, there are certain areas that warrant further attention and elucidation. To improve the quality and significance of the case reports, it is recommended that the authors consider addressing the suggestions outlined below. 

  1. The Introduction could benefit from improved flow and organization. It might be helpful to separate the general information about spasticity and BoNT treatment from the introduction of the cases to ensure a smooth transition between these elements.

REPLY: Following the reviewer’s suggestion, the Introduction has been re-formulated

  1. In the Introduction paragraph mentions that patients experienced "worsening of symptoms" after BoNT treatment, but it does not specify what these symptoms were. Providing a brief overview of the specific symptoms that worsened would make the context clearer.

REPLY: Symptoms refer to dysphagia and dysarthria

  1. While the case reports note potential reasons for the observed adverse effects (e.g., local toxin spread, dosage levels), the discussion lacks an in-depth exploration of the mechanisms underlying these adverse effects. Providing more insights into the pathophysiological basis of the observed symptoms would strengthen the report.

REPLY: The Discussion has been re-formulated following reviewer’s suggestion

  1. The case reports focus on the immediate post-treatment period. Including information about the patients' long-term outcomes and whether any additional treatments or interventions were required for symptom resolution would provide a more comprehensive picture.

REPLY: In order to improve the presentation of our clinical cases, Tables 2 and 3 have been improved including data at discharge.

  1. In the discussion, mention potential alternative treatments or strategies to mitigate the adverse effects of BoNT treatment in patients with severe acquired brain injuries. Are there strategies to avoid or manage these complications

REPLY: As suggested by this reviewer, alternative treatments and strategies have been included in the discussion.   

  1. Discuss the clinical implications of these findings. What can clinicians learn from these cases to improve patient care or make more informed treatment decisions?

REPLY: The Conclusions section has been re-formulated including the clinical algorithm for safer neurotoxic usage of BoNT made by Hristova et al.,  

Round 2

Reviewer 2 Report

Comments and Suggestions for Authors

Dear authors, I greatly appreciated your work in correcting the manuscript in all its critical parts and I believe it is suitable for publication. Excellent work

Author Response

We would like to thank this reviewer for helping us to improve our paper.

All the best

Reviewer 3 Report

Comments and Suggestions for Authors

The authors have provided insightful clinical observations, and the discussion section has been appropriately revised based on previous suggestions. Additionally, the authors have incorporated additional details as requested in the previous version. However, some minor revisions to the formatting are needed to ensure a more cohesive and organized structure. Once these adjustments are made, the manuscript will be ready for publication. The authors are encouraged to carefully address these minor formatting issues for the final enhancement of the manuscript's overall quality.

1.     Please ensure consistency in the reference format throughout the manuscript. It is crucial to maintain a uniform referencing style for improved clarity. Kindly review and revise the references to adhere to a consistent format. Additionally, there are missing DOI details, and there is inconsistency in the inclusion of year and month information. Please address these issues to enhance the overall cohesiveness of the manuscript and ensure a polished presentation, for example refer-1, 3 &7

2.     There are few typo mistakes and formatting which need to be addressed such as the notation “Case No” written as Case n°1, Case n° 2, Case n°3.

3.     In the revised manuscript, it appears that Table 3, Column 1, which originally included the category 'Phonation' in the previous version, now only displays 'II.' The absence of the 'Phonation' label in the current version is noticeable. It would be beneficial to reinstate the 'Phonation' category in Column 1 for clarity and completeness, as it was present in the prior manuscript version.

4.     Please consider renumbering the sections in the manuscript for consistency. The Results section is currently numbered as 3, so it would be appropriate to adjust the Discussion and Conclusion sections to follow sequentially, with Discussion as 4 and Conclusion as 5.

Author Response

    1. Please ensure consistency in the reference format throughout the manuscript. It is crucial to maintain a uniform referencing style for improved clarity. Kindly review and revise the references to adhere to a consistent format. Additionally, there are missing DOI details, and there is inconsistency in the inclusion of year and month information. Please address these issues to enhance the overall cohesiveness of the manuscript and ensure a polished presentation, for example refer-1, 3 &7

    REPLY: DONE

    1. There are few typo mistakes and formatting which need to be addressed such as the notation “Case No” written as Case n°1, Case n° 2, Case n°3.

    REPLY: DONE

    1. In the revised manuscript, it appears that Table 3, Column 1, which originally included the category 'Phonation' in the previous version, now only displays 'II.' The absence of the 'Phonation' label in the current version is noticeable. It would be beneficial to reinstate the 'Phonation' category in Column 1 for clarity and completeness, as it was present in the prior manuscript version.

    REPLY: Dear Reviewer, in Column 1 it appears the category II Phonation. It has never been removed.  

    1. Please consider renumbering the sections in the manuscript for consistency. The Results section is currently numbered as 3, so it would be appropriate to adjust the Discussion and Conclusion sections to follow sequentially, with Discussion as 4 and Conclusion as 5.

    REPLY: DONE